# Entanglement growth from squeezing on the MPS manifold

Sebastian Leontica ✉ & Andrew G. Green

Finding suitable characterizations of quantum chaos is a major challenge in many-body physics, with a central difficulty posed by the linearity of the Schrödinger equation. A possible solution for recovering non-linearity is to project the dynamics onto some variational manifold. The classical chaos induced via this procedure may be used as a signature of quantum chaos in the full Hilbert space. Here, we demonstrate analytically a previously heuristic connection between the Lyapunov spectrum from projection onto the matrix product state (MPS) manifold and the growth of entanglement. This growth occurs by squeezing a localized distribution on the variational manifold. The process qualitatively resembles the Cardy-Calabrese picture, where local perturbations to a moving MPS reference are interpreted as bosonic quasiparticles. Taking careful account of the number of distinct channels for these processes recovers the connection to the Lyapunov spectrum. Our results rigorously establish the physical significance of the projected Lyapunov spectrum, suggesting it as an alternative method of characterizing chaos in quantum many-body systems, one that is manifestly linked to classical chaos.

The emergence of chaos in complex systems is interesting for a variety of reasons. One of the most notable is the effectiveness of statistical principles in its study[1–6], providing an important tool in understanding how macroscopic properties of matter emerge from interactions. While classical chaos is generally well understood, the established tools in the field cannot be straightforwardly applied in the quantum case. The primary difficulty is that quantum mechanics is a linear theory over the full Hilbert space. This implies that there is no manifest way of measuring the divergence of trajectories and therefore spectral tools such as the Lyapunov spectrum are not well-defined. Alternative approaches have been developed to characterize chaos in quantum systems, the most familiar being universal level statistics based on random matrix theory (RMT)[7–11] and the spreading of locally encoded information in the form of out-of-time-ordered-correlators[12–16]. However, these methods also have significant downsides, both from a practical perspective, by being difficult to observe directly in experiments, and from a philosophical perspective, by being apparently dissimilar to their classical counterpart.

A natural route to reconciliation is to take a generalized semiclassical approach by projecting the quantum dynamics onto variational manifolds that capture an increasing fraction of Hilbert space. An understanding of the emergence of chaos in this regime would naturally act as a unifying bridge between its two limiting cases, the fully classical and the fully quantum. In the case of 1D chains with local interactions, the most successful description is provided by matrix product states (MPS)[17–19]. These are the workhorse of many classical algorithms for exploring properties of spin-chains such as the density matrix renormalization group[20–22], time-evolving block decimation[23–25] and time-dependent variational principle (TDVP)[26,27]. Additionally, there is a natural identification between the class of MPS states and the ground states of short-range gapped Hamiltonians, leading to the characterization of possible phases of matter with symmetries in 1D[28,29]. The work of Haegeman et al.[30] showed that these states naturally form a differentiable manifold called the MPS manifold. Furthermore, it is shown to be a Kähler manifold, meaning that the restriction of Schrödinger's equation to the tangent space naturally gives rise to classical Hamiltonian equations of motion (the TDVP equations).

When following the projected dynamics of an initial product state under the action of a local Hamiltonian, the MPS description remains faithful until the entanglement of the quantum state exceeds that

London Centre for Nanotechnology, University College London, London, UK. ✉e-mail: sebastian.leontica.22@ucl.ac.uk

which can be represented on the MPS manifold. At this point the entanglement allowed by the MPS description saturates. It is known that for typical Hamiltonians (excluding exotic phases such as MBL) the growth of entanglement entropy across a cut is asymptotically linear. In conformal field theories, the entanglement growth rate following a quench has been obtained analytically. The mechanism behind it—the Cardy-Calabrese picture[31]—is the production of pairs of excitations with opposite momenta, which lead to entanglement growth as one of the particles travels across the cut. Additionally, work on coupled quantum harmonic oscillators showed that the rate of entanglement growth across a cut is given exactly by the Kolmogorov-Sinai entropy of the associated classical oscillator chain[32,33]. This quantity was introduced as a measure of chaos in deterministic (classical) systems, and was shown to be equal to the sum of the positive Lyapunov exponents, a result known as Pesin's theorem[34].

Here, we show that the growth of entropy, beyond its saturation on the MPS manifold, can also be understood in terms of excitation pairs. Vectors in the tangent space of the manifold correspond to single-tensor perturbations of the reference MPS. Since the MPS have a finite correlation length, these can be interpreted as local excitations[35]. Up to an overcounting factor, these can be treated as quasi-bosonic modes whose classical Lyapunov spectrum determines the entanglement growth rate.

The manuscript is structured as follows: In "The Matrix Product State Manifold" we discuss the matrix product state manifold and its geometry. In "The MPS-Husimi function" we define the MPS-Husimi distribution as a classical representation of some quantum state on the MPS manifold. This is used in "The Entanglement Entropy" to prove that the entropy in its marginals upper bounds the entanglement entropy of the corresponding quantum state. In "Bosonic dynamics from path integral" we construct a path integral over the MPS manifold and show the action can be interpreted as quasi-bosonic. The resulting equation is revisited in "Variational principle in second tangent space", where a variational principle is used to show and correct an overcounting error incurred due to naive application of the path integral method. Finally, we summarize our results and discuss potential future work in "Variational principle in second tangent space". A schematic representation of the connection between entanglement growth and squeezing is given in Fig. 1.

## Results

### The matrix product state manifold

In this section we give a brief overview of the MPS ansatz, highlighting the main ideas and tools that are used in the rest of the manuscript. The reader familiar with tensor network methods may skip this section and refer back to it when needed.

We consider a linear chain $\mathcal{N}$ composed of $N$ qudits with local dimension $d$, whose Hilbert space is spanned by pure quantum states $|\psi\rangle \in \mathbb{C}^{d^N}$. If we denote by $|n_1 n_2 \ldots n_N\rangle$ a basis state of the many-body Hilbert state, with each $n_i$ running from 1 to $d$, then an MPS is simply a linear combination of the basis states, with coefficients given by products of matrices. For the case of open boundary conditions, which we assume throughout the work, it is generally represented as

$$\Psi[\mathcal{A}] = \sum_{\{n_i\}} A_{n_1}^{[1]} A_{n_2}^{[2]} \ldots A_{n_N}^{[N]} |n_1 n_2 \ldots n_N\rangle,$$

$$= \quad \text{(1)}$$

where $A_{n_i}^{[i]}$ is a $D_{i-1} \times D_i$ matrix with complex entries for each site $i$ and physical index $n_i$. We can collect all $d$ matrices located at some site $i$ into a single rank 3 tensor $A^{[i]}$. The quantities $D_i$, for $i$ running from 0 to $N$, are called virtual bond dimensions and they restrict the amount of

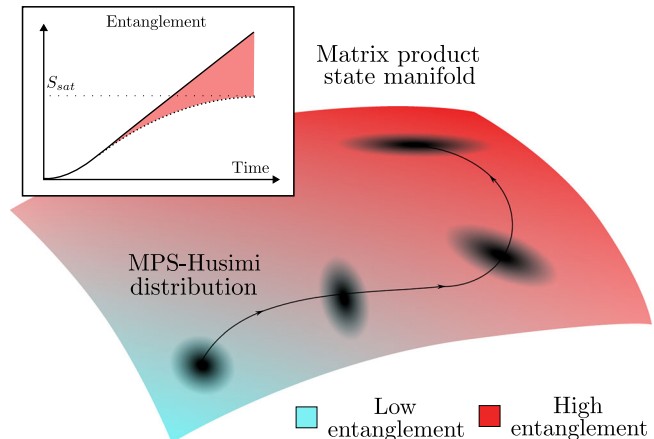

**Fig. 1 | Schematic representation of the entanglement growth during a variational evolution on the matrix product state manifold.** Due to overcompleteness, an arbitrary state can be represented as a wavepacket on the variational manifold, a unique localized construction given by the MPS-Husimi distribution introduced in the main text. A weakly entangled state can be represented by a tightly localized wavepacket starting in the low entanglement corner of the MPS manifold. Initially this distribution evolves undistorted, in a manner determined by the time-dependent variational principle. Above the entanglement saturation limit of the manifold, the wavepacket becomes squeezed. Excess entanglement is recovered from the classical correlations (squeezing) of the associated local MPS-Husimi distribution (light-red contribution in inset).

entanglement the state can have across a cut between sites $i$ and $i+1$. Since the state is pure, there can be no entanglement across a cut located at either end, so we must have $D_0 = D_L = 1$. For technical reasons, we will also assume that $dD_{i-1}/D_i \in \mathbb{N}^+$ and $D_i \le \min(d^i, d^{L-i})$. It is known that the parameterization given by the set of rank 3 tensors $\mathcal{A} = \{A^{[i]}\}_{i \in \mathcal{N}}$ does not uniquely identify a many-body state $\Psi[\mathcal{A}]$. This gauge freedom can be fixed by imposing additional restrictions on the local tensors. One of the most common solution is to assume the tensor is in left canonical form, which is equivalent to saying it can be embedded in a unitary matrix as

$$\quad = \quad , \quad \text{(2)}$$

where the ingoing and outgoing arrows denote the input and output legs of the unitary $U^{[i]} \in \text{SU}(dD_{i-1})$. For the operation to be unitary, the dimensions of the inputs and outputs need to match, so the top leg must correspond to a Hilbert space of dimension $dD_{i-1}/D_i$. In this gauge, the tensors satisfy the equation

$$\quad = \quad , \quad \text{(3)}$$

which can also be used iteratively to prove the normalization condition $\Psi[\mathcal{A}]^\dagger \Psi[\mathcal{A}] = 1$. We define the right-environments starting from the right end of the chain $\Gamma_N = 1$ according to the recurrence

$$\quad = \quad \Gamma_{i-1}. \quad \text{(4)}$$

While this removes some of the redundancy, the many-body state $\Psi[\mathcal{A}]$ is still not uniquely defined by the set $\mathcal{A}$. Any element $(U_1, U_2, \ldots U_{N-1}) \in SU(D_1) \times SU(D_2) \times \cdots \times SU(D_{N-1}) : = G_\mathcal{N}$, along with $U_0 = U_N = 1$, acting as $\mathcal{A} \to \mathcal{A}' = \{U_{i-1}^\dagger A^{[i]} U_i\}_{i=1}^N$ will produce the same state $\Psi[\mathcal{A}'] = \Psi[\mathcal{A}]$. Additionally, global phases do not affect observable properties of the state, so each site has an additional $U(1)$ gauge freedom under $A_i \to e^{i\phi} A_i$. We can use the group actions defined in this way to define an equivalence relation between sets $\mathcal{A} \sim \mathcal{A}'$. We call the manifold of orbits under the previously defined group actions by $\mathcal{M}$, the MPS manifold. The geometry of this manifold has been thoroughly investigated by Haegeman et al.[35] and was shown to have the structure of a Kähler manifold.

A non-redundant local parameterization in the vicinity of some reference MPS[26,36] can be constructed by the right contraction of the unitary $U$ with a parameterized unitary $W(x) = \exp\left(x_i \langle 0| \otimes \Gamma_i^{1/2} - |0\rangle \otimes \Gamma_i^{1/2} x_i^\dagger\right)$. The site-dependent parameters $x_i$ are organized into an arbitrary complex 3-tensor, with the only condition that it must be null when the physical index is set to 0. The contractions in the exponent are made more clear in the diagrammatic form

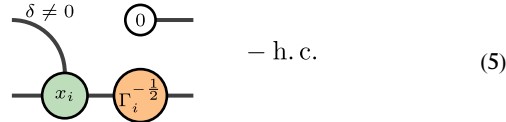

$$- \, h.c. \tag{5}$$

With these ingredients, the updated MPS tensor is given by

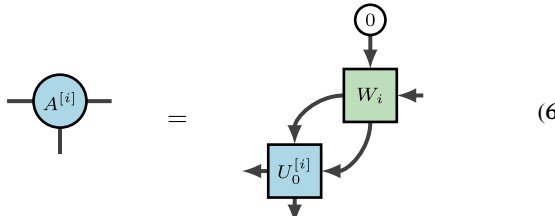

$$\tag{6}$$

This is equivalent to the Thouless' theorem of ref. 37

$$A^{[i]}(x) = e^{B^{[i]}(x) A_0^{[i]\dagger} - A_0^{[i]} B^{[i]\dagger}(x)} A_0^{[i]}, \tag{7}$$

where the 3-tensor $A^{[i]}$ is turned into a matrix by combining the physical and the left bond indices. $B^{[i]}(x)$ is another 3-tensor (turned into a matrix via the same convention) given by

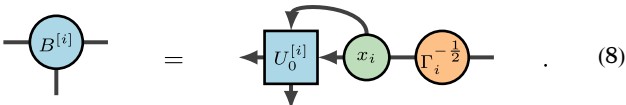

$$\tag{8}$$

The condition that $x_i$ is null when its physical index is zero ensures that $A^{[i]\dagger} B^{[i]} = 0$ for multiplications in matrix form.

The parameterization $\Psi(x)$ with $x = \{x_i\}_{i=1}^N$ is surjective and the derivatives along components of the 3-tensors $x_i$ form an orthonormal basis of the MPS tangent space at $\Psi_0$. If we let $\mu$ be an index for the entries of x at some site $i$, then the tangent vectors are given by

$$\partial_\mu^{(i)} \Psi = \sum_{\{n_i\}} A_{n_1}^{[1]} A_{n_2}^{[2]} \ldots B_{\mu, n_i}^{[i]} \ldots A_{n_N}^{[N]} |n_1 n_2 \ldots n_N\rangle, \tag{9}$$

where $B_\mu^{(i)} = \partial_\mu B^{[i]} = \partial_\mu A^{[i]}|_{x=0}$ is the derivative of $A^{[i]}(x)$ along the corresponding entry in $x_i$. This definition ensures that $\Psi^\dagger \partial_\mu^{(i)} \Psi = 0$ and $\partial_\mu^{(i)} \Psi^\dagger \partial_\nu^{(j)} \Psi = \delta_{ij} \delta_{\mu\nu}$.

A measure on the MPS manifold can be obtained starting from the Haar measure of the unitary group. If $f(\Psi)$ is some function defined on the MPS manifold, we can define its integral via the following equation

$$\int_\mathcal{M} d\Psi f(\Psi) = \int_{Haar} \prod_{i-1}^N dU^{[i]} f(\Psi[\mathcal{A}]), \tag{10}$$

with the local tensors defined in terms of the unitaries as in Eq. (2). The Haar integrals can then be performed using known results such as the Weingarten calculus[38].

The construction above can be extended in a straightforward way to discuss a particular segment of an MPS, rather than the full chain. If we let $\mathcal{I}$ represent some segment stretching between bond indices $e_L$ and $e_R$, then we can write down a segment MPS

$$\Psi_\mathcal{I}[\mathcal{A}_\mathcal{I}] = \frac{1}{\sqrt{D_{e_R}}} \sum_{\{n_i\}_{i \in \mathcal{I}}} \overrightarrow{\prod_{i \in \mathcal{I}}} A_{n_i}^{[i]} |\{n_i\}_{i \in \mathcal{I}}\rangle,$$

$$= \frac{1}{\sqrt{D_{e_R}}} \quad e_L \!-\! \langle A^{[L]} \rangle \!-\! \cdots \!-\! \langle A^{[R]} \rangle \!-\! e_R \tag{11}$$

where the bond indices at the edges are free and can be interpreted as physical. We will refer to these additional sites as edge ancillae. By analogy with the previous construction for the full chain, we will denote the manifold of different states that can be obtained in this way by $\mathcal{M}_\mathcal{I}$. If $\mathcal{I}$ extends to one end of the chain and $\mathcal{M}_{\mathcal{N}\backslash\mathcal{I}}$ is the manifold of MPS segments over the rest of the chain, then we can recover the manifold over the entire chain $(\Psi_\mathcal{I}, \Psi_{\mathcal{N}\backslash\mathcal{I}}) \to \Psi$ by contracting over the common edge and quoting out the resulting gauge freedom.

Having established the main terminology and notation, we can move on to discuss methods for representing general quantum states on the manifold.

**The MPS-Husimi function**

In this section we aim to construct a classical representation of arbitrary quantum states via projection onto the MPS manifold. The projection mechanism can be understood as a form of measurement onto the over-complete basis of MPS. The probability of collapsing onto some MPS is given by its overlap with the original quantum state, which is the desired classical distribution representing the state on the manifold. This idea can easily be extended to representing reduced states on some subset of all sites by measuring in a basis formed from the segment MPS defined in Eq. (11). This procedure leads us to consider the following definition of the MPS-Husimi distribution

$$Q_\mathcal{I}(\Psi_\mathcal{I}) = \Psi_\mathcal{I}^\dagger \rho_\mathcal{I} \Psi_\mathcal{I},$$

$$= \frac{1}{D_{e_R}} \left( \begin{array}{c} A^{[l]} \cdots A^{[r]} \\ \boxed{\rho_\mathcal{I}} \\ \overline{A}^{[l]} \cdots \overline{A}^{[r]} \end{array} \right), \tag{12}$$

where $\rho_\mathcal{I}$ is the reduced density matrix of the state we wish to represent onto sites in $\mathcal{I}$. This is similar to the definition of the Husimi distribution used in quantum optics[39,40]. Since it will be larger in regions of the manifold that better resemble the initial state, it is natural to interpret this as a classical representation of the state onto the MPS manifold.

We prove that this distribution forms a proper probability distribution of a quantum measurement onto the MPS manifold, with the

details given in the Supplementary Note 1. The result is an ensemble of reduced density matrices of MPS states, with a weight given by the MPS-Husimi function. More formally, we prove that the linear function transforming reduced density matrices given by

$$\mathcal{E}(\rho_{\mathcal{I}}) = \int_{\mathcal{M}_{\mathcal{I}}} \frac{d\Psi_{\mathcal{I}}}{\mathcal{V}_{\mathcal{I}}} Q_{\mathcal{I}}(\Psi_{\mathcal{I}}) \mathrm{Tr}_E(\Psi_{\mathcal{I}} \Psi_{\mathcal{I}}^\dagger),$$ (13)

where $\mathrm{Tr}_E$ means tracing over the edge ancillae, forms a good quantum channel[41–45].

We will outline some of the most important properties of the MPS-Husimi distribution. Firstly, in order to be a valid distribution we must show that it is positive semi-definite and normalized. The first property can easily be checked from its definition in Eq. (12), using the assumed positive semi-definiteness of the density matrix $\rho_{\mathcal{I}}$. The normalization follows from the resolution of identity over the set of segment MPS via

$$\int_{\mathcal{M}_{\mathcal{I}}} \frac{d\Psi_{\mathcal{I}}}{\mathcal{V}_{\mathcal{I}}} Q_{\mathcal{I}}(\Psi_{\mathcal{I}}) = \int_{\mathcal{M}_{\mathcal{I}}} \frac{d\Psi_{\mathcal{I}}}{\mathcal{V}_{\mathcal{I}}} \Psi_{\mathcal{I}}^\dagger \rho_{\mathcal{I}} \Psi_{\mathcal{I}} = \mathrm{Tr}\rho_{\mathcal{I}} = 1.$$ (14)

Another question that naturally arises is whether there is any relationship between the MPS-Husimi distributions of different segments. The answer is that they are the marginals of the full MPS-Husimi distribution, in the following sense

$$Q_{\mathcal{I}}(\Psi_{\mathcal{I}}) = \int \frac{d\Psi_{\mathcal{N}\setminus\mathcal{I}}}{\mathcal{V}_{\mathcal{N}\setminus\mathcal{I}}} Q(\Psi_{\mathcal{N}\setminus\mathcal{I}} \cdot \Psi_{\mathcal{I}}),$$ (15)

where $\Psi_{\mathcal{N}\setminus\mathcal{I}} \cdot \Psi_{\mathcal{I}}$ denotes contraction over the edge ancillae.

## The entanglement entropy

In this section, we will use the machinery introduced so far to understand the entanglement properties of the arbitrary state $\rho$ from the classical distribution it induces on the MPS manifold. The result is that the entanglement is upper bound by a combination of average intrinsic entanglement of MPS in the distribution and classical correlations between parameters of the distribution itself.

The most common metric used to discuss bipartite entanglement properties of pure many-body states $\rho = |\psi\rangle\langle\psi|$ is the von Neumann entanglement entropy[46] defined by

$$S_{vN}(\rho_{\mathcal{I}}) = -\mathrm{Tr}(\rho_{\mathcal{I}} \ln \rho_{\mathcal{I}}),$$ (16)

where $\rho_{\mathcal{I}} = \mathrm{Tr}_{\mathcal{N}\setminus\mathcal{I}}\rho$ is again the reduced density matrix. Several extensions based on this quantity exist for the case of mixed states and multi-partite information[47], but we do not discuss them here.

States that can be represented exactly as points on the MPS manifold have intrinsic entanglement entropy. For the simplest case where the sub-regions are separated by a single cut ($\mathcal{I}$ extends all the way to the left end of the chain) it can be shown that the entanglement entropy is given by

$$S_{vN}(\rho_{\mathcal{I}}) = -\mathrm{Tr}(\Gamma_{e_R} \ln \Gamma_{e_R}),$$ (17)

where $\Gamma_{e_R}$ is the right environment defined by Eq. (4) at the right edge of the segment. Since $\Gamma_{e_R}$ is a $D_{e_R} \times D_{e_R}$ positive semi-definite matrix of unit trace, its entanglement entropy is bounded by $\ln D_{e_R}$. Due to this limitation, we expect states with higher entanglement entropy to be difficult to approximate by single points on the manifold. A reasonable attempt for a way forward in this situation is to consider wavepackets on the manifold, instead of single points. It is easy to see that any state can be exactly represented (non-uniquely) as a wavefunction on the manifold, owing to the overcompleteness of the MPS states. It should be intuitively clear that the minimum support necessary for an accurate

representation must be somehow related to excess entanglement, since all states of entanglement below saturation can be represented as points, but it has proven difficult to establish a direct connection.

The right way forward is instead to look at the MPS-Husimi distribution of the state on the manifold. In particular, the crucial thing to note is that the channel defined in Eq. (13) is unital, meaning that it has the identity as a stationary point $\mathcal{E}(I) = I$. Such channels are known to always increase entropy[48], so we must have

$$S(\rho_{\mathcal{I}}) \leq S(\mathcal{E}(\rho_{\mathcal{I}})).$$ (18)

Moreover, since we have shown that $Q_{\mathcal{I}}$ is a proper distribution, the density matrix $\rho_{\mathcal{I}}'$ obtained after sending the state through the quantum channel is a convex combination (Eq. (13)) of the MPS reduced density matrices $\rho_{\mathcal{I}}(\Psi_{\mathcal{I}}) = \mathrm{Tr}_E(\Psi_{\mathcal{I}}\Psi_{\mathcal{I}}^\dagger)$. For the entropy of such states the following bound is known

$$S(\rho_{\mathcal{I}}') \leq \int_{\mathcal{M}_{\mathcal{I}}} \frac{d\Psi_{\mathcal{I}}}{\mathcal{V}_{\mathcal{I}}} Q_{\mathcal{I}}(\Psi_{\mathcal{I}}) S(\rho(\Psi_{\mathcal{I}})) + S_W(Q_{\mathcal{I}}),$$ (19)

where $S_W(Q_{\mathcal{I}})$ is the Wehrl (or classical) entropy of the MPS-Husimi distribution

$$S_W(Q_{\mathcal{I}}) = -\int_{\mathcal{M}_{\mathcal{I}}} \frac{d\Psi_{\mathcal{I}}}{\mathcal{V}_{\mathcal{I}}} Q_{\mathcal{I}}(\Psi_{\mathcal{I}}) \ln Q_{\mathcal{I}}(\Psi_{\mathcal{I}}).$$ (20)

If we combine the two inequalities we get the desired relation between the entanglement entropy of some state and the support of its representation on the MPS manifold

$$S(\rho_{\mathcal{I}}) \leq \langle S(\rho(\Psi_{\mathcal{I}})) \rangle_{Q_{\mathcal{I}}} + S_W(Q_{\mathcal{I}}),$$ (21)

where $\langle \cdot \rangle_{Q_{\mathcal{I}}}$ is a shorthand for the weighted average in Eq. (19).

## Bosonic dynamics from path integral

The bound we obtain has a clear and intuitive interpretation. The first term is simply the average entropy of the matrices $\rho(\Psi_{\mathcal{I}})$. Since this can be understood as the entanglement entropy of a segment of a single MPS, it is upper-bounded by $\log D_{e_L} D_{e_R}$. In the initial stage of a quench, when a single MPS is sufficiently accurate to represent the state, the MPS-Husimi distribution can be imagined as a tightly-packed Gaussian. In this case, the Wehrl entropy only contributes a small constant, so the first term is the main component in the bound. As the entropy reaches the saturation limit of the manifold, we expect to enter a classically chaotic region and observe a squeezing of the distribution along the classical Lyapunov basis directions. These generally introduce classical correlations between different parts of the chain, so the entropy of the marginal distribution grows. The growth is linear for typical systems, at a rate related to the classical Kolmogorov-Sinai entropy of the region. This is consistent with numerical observations in ref. 49 and exact analytic results obtained in free bosonic systems[32].

Since the total probability is a conserved quantity, the evolution of the Husimi Q-function must be governed by a continuity equation

$$\frac{\partial Q}{\partial t} + \nabla \cdot J = 0,$$ (22)

where $J$ is a probability current that depends on the local value of $Q$ and its derivatives. Deriving the functional form of the current for the MPS-Husimi function is a difficult task in general, since we cannot neglect the curvature of the manifold. We start by taking the time derivative of Eq. (12), with $\mathcal{I}$ taken to be the entire chain, and use the von Neumann equation for the density matrix to get

$$\frac{\partial Q(\Psi)}{\partial t} = -i\Psi^\dagger[\mathcal{H}, \rho]\Psi.$$ (23)

A perturbative expansion around a reference MPS $\Psi_0$ can be performed to rewrite the RHS in terms of derivatives of $Q$ at $\Psi_0$. While this method is exact, it produces complicated equations and does not provide much intuition for the underlying process of entanglement production. Instead, we will pursue an analogy with free bosons, where the link between entanglement production and growth of phase space volumes has been thoroughly investigated[32]. The MPS path integral formulation provides a very natural connection, albeit requiring some correction that we will discuss in "Variational principle in second tangent space". The details of the construction were previously given in ref. 36, but we reproduce the main ideas here.

Let us start by considering the propagator of $\mathcal{H}$ and proceed in the usual way by introducing resolutions of identity at small time-intervals $\Delta t$

$$\mathcal{U}(t,0) = e^{-i\mathcal{H}t} = \int D\Psi \prod_n \left( e^{-i\mathcal{H}\Delta t} \Psi_n \Psi_n^\dagger \right)$$
$$= \int D\Psi \exp\left[ iS_B[\Psi,\Psi^\dagger] - i\int_0^t d\tau \Psi^\dagger(\tau)\mathcal{H}\Psi(\tau) \right], \quad (24)$$

where $S_B$ is the geometric Berry phase, whose form we will discuss later, and $D\Psi$ is a product measure coming from the resolutions of identity. As is common in path integral treatments, we have assumed that the majority of the contribution is coming from continuous paths on the MPS manifold. For it to be computationally useful, the action needs to be expressed using some explicit parameterization. In this work, we will choose the local parameterization in Eq. (7). In the local coordinate system, we can express the propagator as the path integral

$$\int Dx D\overline{x} \exp\left[ iS_B[x,\overline{x}] - i\int_0^t d\tau \mathcal{H}(x(\tau),\overline{x}(\tau)) \right], \quad (25)$$

where we introduced the classical Hamiltonian $\mathcal{H}(x,\overline{x}) = \Psi^\dagger(x)\mathcal{H}\Psi(x)$.

The most straightforward way to deduce the form of the Berry phase functional is requiring that the minimization condition for the action recovers the TDVP equations. With this condition in mind we find that

$$S_B[x,\overline{x}] = \frac{i}{2}\sum_i \int_0^t d\tau \text{Tr}\left( \frac{dx_i}{d\tau} x_i^\dagger - \frac{dx_i^\dagger}{d\tau} x_i \right). \quad (26)$$

If we index the independent tangent directions of the manifold at each site by $\mu$, we can write $x = \sum_\mu x_\mu e_\mu$, where $x_\mu$ is some complex number and $e_\mu$ is a 3-tensor with a single non-zero entry equal to 1. With this convention we obtain the orthonormality condition $\text{Tr} e_\mu^\dagger e_\nu = \delta_{\mu\nu}$. Using this, it is easy to check that the Berry phase decomposes into a sum of geometric phases corresponding to each mode. The action of the problem is then

$$S[x,\overline{x}] = \int_0^t d\tau \left\{ \frac{i}{2}\left( \sum_{i,\mu} \frac{dx_{i\mu}}{d\tau} \overline{x}_{i\mu} - \frac{d\overline{x}_{i\mu}}{d\tau} x_{i\mu} \right) - H \right\}. \quad (27)$$

Expressing the propagator in this form allows for an important insight, which is that the same action would be obtained for a collection of bosonic modes, when resolved in the over-complete set of coherent states. This suggests that we can produce an approximate solvable model of the dynamics of wavepackets on the MPS manifold by expanding the Hamiltonian to quadratic order and exchanging the complex variables $x_{i\mu}, \overline{x}_{i\mu}$ for independent bosonic creation and annihilation operators $a_{i\mu}, a_{i\mu}^\dagger$. Explicit calculations of the expanded Hamiltonian are given in ref. 50, but its exact form is not important for the present discussion. It is sufficient to know that this Hamiltonian is quadratic and contains local anomalous terms of the form $a_{i,\mu}^\dagger a_{j,\nu}^\dagger$ for $i,j$ close to each other, which produce excitation pairs.

We have so far shown that the growth of correlations between variables at different sites leads to entanglement, and that this growth of correlations can be modeled via a system of free bosons. In such systems, the rate of growth of the entanglement entropy of some subset of the chain is equal to the subsystem exponent. The latter is a classical measure of chaos that can be obtained from the Lyapunov exponents, and when the subsystem is large enough it is simply equal to the Kolmogorov-Sinai entropy. A formal definition of the subsystem exponent $\Lambda_\mathcal{I}$ can be defined by considering the evolution of an infinitesimally small phase space cube $v$. If $\text{vol}_\mathcal{I} v$ is the volume of the projection of $v$ onto a hyperplane spanned by infinitesimal variations of the $\vec{x}_\mathcal{I}$ coordinates corresponding to segment $\mathcal{I}$, then the subsystem exponent is defined as the long-time limit

$$\Lambda_\mathcal{I} = \lim_{t\to\infty} \frac{1}{t}\log \frac{\text{vol}_\mathcal{I} \Phi_t(v)}{\text{vol}_\mathcal{I} v}. \quad (28)$$

Similar quantities relating to the growth of phase space volumes have been thoroughly investigated in the classical chaos literature. There are also efficient algorithms that can compute them for arbitrary dynamics on manifolds[49].

## Variational principle in second tangent space

The path integral method outlined in the previous section has its merits for suggesting an important analogy between the local derivatives of the MPS manifold and quantum bosonic modes. However, in its current form, the prediction overestimates the rate of growth of entanglement by a factor of $D^2$[49]. Here, we show that this effect is geometric in nature and should appear, after a more careful treatment, as a coupling between modes on different sites in the Berry phase.

We have shown how to construct an orthonormal basis for the tangent space in Eq. (9). The vectors spanning higher order tangent spaces are constructed similarly by taking higher derivatives in coordinates at different sites. Traditional TDVP techniques generate dynamics on the MPS manifold by projecting the effects of the Hamiltonian into the local tangent space. Using this technique, the quantum state is simply transported across the manifold and the effects of squeezing are not observed. Looking at the squeezing of classical distributions following the TDVP equation is a good starting point, but this leads to the overcounting effect mentioned early when applied to the MPS-Husimi distribution. In order to probe this higher order effect we introduce a variational ansatz that also includes possible contributions from the second tangent space. At some point $x$ on the manifold this can be explicitly written as

$$\widetilde{\Psi}_0(\alpha) = \Psi[\mathcal{A}_0] + \sum_{i<j,\,\mu\nu} \alpha_{\mu\nu}^{(i,j)} \partial_\mu^{(i)} \partial_\nu^{(j)} \Psi[\mathcal{A}_0], \quad (29)$$

where the $\alpha_{\mu\nu(i,j)}$ are also variational parameters. We can construct a variational principle for the evolution of this state and show that we obtain various contributions as we transport the second tangent space around the manifold. The details of this are shown in the Supplementary Note 2. The resulting equation is very complex in the most general case, so we look for a simplified scenario in order to get a qualitative understanding of the squeezing. We focus on the fixed points of the Hamiltonian on the manifold, where $\Delta x = 0$ is a solution. In this case we see that the evolution of $\alpha$ must obey the set of equations

$$\left( \partial_\eta^{(x)} \partial_\delta^{(y)} \Psi \right)^\dagger \left[ -i\mathcal{H}\widetilde{\Psi}_0(\alpha) - \sum_{i<j,\,\mu\nu} \dot{\alpha}_{\mu\nu}^{(i,j)} \partial_\mu^{(i)} \partial_\nu^{(j)} \Psi \right] = 0, \quad (30)$$

for all values of $\eta, \delta, x, y$. Solving this set of equations is not as straightforward as in the simple TDVP case because the second tangent space Grammian

$$G_{\eta\delta,\,\mu\nu}^{(xy,\,ij)} = \left( \partial_\eta^{(x)} \partial_\delta^{(y)} \Psi \right)^\dagger \partial_\mu^{(i)} \partial_\nu^{(j)} \Psi, \quad (31)$$

may be degenerate. Let us first consider the action of the Hamiltonian on vectors $\partial_\mu^{(i)}\partial_\nu^{(j)}\Psi$ in the second tangent space. We would like to find the best approximation for this action by a free propagation of the derivatives. Then our goal is to find a matrix $\epsilon$ such that

$$\mathcal{H}\partial_\mu^{(i)}\partial_\nu^{(j)}\Psi \sim \sum_{\eta,x}\left(\epsilon_{\eta\mu}^{(xi)}\partial_\eta^{(x)}\partial_\nu^{(j)}\Psi + \epsilon_{\eta\nu}^{(xj)}\partial_\mu^{(i)}\partial_\eta^{(x)}\Psi\right). \tag{32}$$

We can again proceed by minimizing the norm of the difference between the two sides, using $\epsilon$ as a variational parameter. We choose to take a shortcut to this procedure and note that, for large $N$, the norm is dominated by contributions from vectors with excitations spread apart $j - i \gg \xi$, where $\xi$ is the correlation length of the reference MPS $\mathcal{A}_0$. In this regime it can be shown that the Grammian is approximately

$$G_{\eta\delta,\mu\nu}^{(xy,ij)} = \delta_{\eta\mu}\delta_{\delta\nu}\delta_{xi}\delta_{yj} + \mathcal{O}(e^{-|j-i|/\xi}), \tag{33}$$

so the problem is equivalent to finding the best solution to the single particle evolution

$$\mathcal{H}\partial_\mu^{(i)}\Psi \sim \sum_{\nu,j}\epsilon_{\nu\mu}^{(ji)}\partial_\nu^{(j)}\Psi. \tag{34}$$

Since the space of single excitations is orthonormal, the solution to this is simply

$$\epsilon_{\mu\nu}^{(ij)} = \left(\partial_\mu^{(i)}\Psi\right)^\dagger \mathcal{H}\partial_\nu^{(j)}\Psi. \tag{35}$$

Let us now consider the part of the Hamiltonian that creates pairs of excitations from the vacuum

$$\mathcal{H}\Psi \sim \sum_{\mu\nu,ij}\Delta_{\mu\nu}^{(ij)}\partial_\mu^{(i)}\partial_\nu^{(j)}\Psi, \tag{36}$$

the MPS path-integral method outlined in "Bosonic dynamics from path integral" suggests that $\Delta$ is simply obtained from an overlap, the same as in the previous equation. This is, however, not accurate, as in this case the Hamiltonian will tend to produce pairs of excitation on nearby sites $j - i \sim \xi$ where the Grammian is highly degenerate. The simplest way of treating the degeneracy is to find a resolution of identity over the second tangent space. Finding an exact resolution of identity in this space is difficult, since the degeneracy of the Grammian is only approximate. Instead, we will attempt to find an approximate resolution of the identity of the form

$$\sum_{\mu\nu,i<j}\frac{1}{\rho(|j-i|)}\partial_\mu^{(i)}\partial_\nu^{(j)}\Psi\left(\partial_\mu^{(i)}\partial_\nu^{(j)}\Psi\right)^\dagger \sim \mathbb{1}_{(2)}, \tag{37}$$

where $\mathbb{1}_{(2)}$ represents identity over the second tangent space and $\rho$ is a density of states depending only on the distance between the quasi-particles. In the Supplementary Note 3 we show that the density of states behaves like

$$\rho(x) \sim 1 + (D^2 - 1)e^{-x/\xi}. \tag{38}$$

Multiplying Eq. (30) by $\rho^{-1}(|y - x|)\partial_\eta^{(x)}\partial_\delta^{(y)}\Psi$ and summing over all free indices we see that a possible solution of the equation is

$$\dot{\alpha}_{\mu\nu}^{(i,j)} = -i\sum_{\eta,x}\left(\epsilon_{\mu\eta}^{ix}\alpha_{\eta\nu}^{xj} + \epsilon_{\nu\eta}^{jx}\alpha_{\mu\eta}^{ix}\right)$$
$$-i\frac{1}{\rho(|j-i|)}\left(\partial_\mu^{(i)}\partial_\nu^{(j)}\Psi\right)^\dagger \mathcal{H}\Psi, \tag{39}$$

such that the anomalous terms are given by

$$\Delta_{\mu\nu}^{(ij)} = \frac{1}{\rho(|j-i|)}\left(\partial_\mu^{(i)}\partial_\nu^{(j)}\Psi\right)^\dagger \mathcal{H}\Psi. \tag{40}$$

We now recognize that Eq. (39) can be obtained in a bosonic system driven by the free Hamiltonian

$$\mathcal{H}_B = E_0 + \sum_{\mu\nu,ij}\left(\epsilon_{\mu\nu}^{(ij)} - \delta_{\mu\nu}\delta_{ij}E_0\right)a_\mu^{(i)\dagger}a_\nu^{(j)}$$
$$+ \sum_{\mu\nu,i<j}\left(\Delta_{\mu\nu}^{(ij)}a_\mu^{(i)\dagger}a_\nu^{(j)\dagger} + \overline{\Delta}_{\mu\nu}^{(ij)}a_\mu^{(i)}a_\nu^{(j)}\right), \tag{41}$$

where $E_0 = \Psi^\dagger\mathcal{H}\Psi$. The linear rate of growth of the entanglement entropy across a cut in a bosonic system coincides with the classical growth rate of subsystem entropy discussed in "The Entanglement Entropy". However, the anomalous terms we obtain using the variational method differ from those found in the classical non-linearities by a factor of $1/D^2$ for local Hamiltonians. Since the production of entangled pairs in the classical system is overestimated, we expect the Kolmogorov-Sinai entropy of the classically projected dynamics to be larger by the same factor of $D^2$. This has indeed been found in recent numerical simulations[49].

## Discussion

The results presented in this paper provide a solid theoretical foundation for the recently discovered connection between entanglement dynamics in local many-body systems and the classical TDVP equations obtained via a projection onto the entangled MPS manifold[49,50]. The Kolmogorov-Sinai entropy, a measure of chaos in classical systems that can be computed from the divergence rates of trajectories, is established as an upper bound on the rate of entanglement growth, up to a factor of $D^2$ owing to an over-counting of degrees of freedom. This connection establishes the Lyapunov spectrum of the projected dynamics as a physically meaningful tool for characterizing chaos in many-body systems.

We associate to any quantum state a classical distribution on the manifold, dubbed the MPS-Husimi function due to its similarity to the Husimi function in quantum optics[39]. Then, we prove that the entanglement entropy of the quantum state is upper bound by a sum of two distribution-dependent terms. The first represents the average intrinsic entanglement entropy of the MPS making up the distribution, and the other an additional extrinsic entropy owing to classical correlations in the parameters of the distribution itself.

When the quantum state in question is on the manifold, the distribution can be pictured as a tight, symmetric Gaussian surrounding it. Before saturating the entanglement, the projected equations are linear and the distribution captures entanglement growth only via the first term, by simply shifting its center towards higher entanglement parts of the manifold. When saturation is reached, the first term is insufficient and further growth in entanglement manifests itself through the onset of classical correlations in the distribution, i.e., by squeezing the Gaussian. We note that classical flows similar in spirit to the ones we discuss can occur organically in systems with some form of local monitoring that is strong enough to prevent the evolution towards volume-law entanglement.

This indicates a possible connection to non-linear classical equations of motion, as they would produce a similar squeezing of the distribution. This is subtle, however, as it is not a priori clear that the MPS-Husimi function should follow the classical flow equations of a real distribution. The MPS path integral method reveals that the true dynamics can be understood as a system of free bosons, whose quadratic Hamiltonian can be easily constructed. This analogy suggests a rate of squeezing in line with that of distributions obeying the classical equations of motion. However, a more careful variational principle reveals that the local squeezing of the MPS-Husimi function should be a factor of $1/D^2$ smaller than that of a classical distribution, due to an overcounting of the number of independent excitations above the MPS vacuum − a fact that was discovered heuristically in previous numerical works[49] and obtained a theoretical foundation here.

In the expansion of the Hamiltonian, we neglect terms at higher than second order. The justification for this is that it is the simplest truncation that captures the qualitative quasi-particle production mechanism of entanglement growth. The equation governing the Husimi function Eq. (22) is of second order in spatial derivatives, and the global features are captured through a position dependence in the coefficients, such that parts of the Husimi function in regions of the manifold with small overlap are squeezed independently of each other. Higher order terms act as small interaction terms in the bosonic picture, which may lead to a dressing of the quasi-local excitations, without meaningfully affecting the physical picture. In cases where this approximation leads to significant numerical inaccuracy, the quadratic approximation is nevertheless a good starting point towards perturbative corrections.

An interesting avenue for further investigation is the interpretation of our results for the projected dynamics of quantum integrable systems. In certain situations, such as the stable regime of coupled quantum harmonic oscillators discussed in ref. 51, the entanglement entropy was shown to be periodic in time. This is consistent with the projected equations being classically integrable. However, for quantum integrable systems that show a linear growth of entanglement entropy, our results imply that all classical projections must be chaotic, at least in some parts of their phase space. It would be very interesting to check how the projected flows of such systems differ from those of a chaotic quantum system. This would require a more exact method for treating the degeneracy discussed in "Variational principle in second tangent space", which we plan to develop in the future. As an interesting situation on the opposite side of the spectrum, some non-integrable quantum systems were found to have regions of phase space with quasi-periodic orbits[52] when projected onto the MPS manifold. As expected in light of our result, initializing the quench in these regions leads to a strongly suppressed entanglement growth rate.

Our work is an important step towards understanding the connection between quantum chaos and its classical counterpart. Statistical mechanics has lead to great insight in many areas of physics, so the prospect of linking entanglement properties of matter to the behavior of classical distributions on some manifold may prove very powerful. An interesting question for future work is whether one can incorporate the over-counting of local degrees of freedom directly into the path integral representation, leading to accurate fluctuation corrections to MPS. This would further expand the computational power of MPS by producing accurate results at lower bond dimensions. A different development may be to study how local measurements affect the MPS-Husimi distribution, potentially elucidating the nature of entanglement and purification transitions[53–55] through modifications in the Lyapunov spectrum. Reframing characterization of quantum chaos in a manner closely aligned with its classical counterpart opens a new window upon quantum chaos and suggests a way forward in many problems of topical interest. We hope this new direction will prove fruitful in future projects.

## Data availability

All information necessary to support the findings of this study is available in the main article and Supplementary Information file.

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

## Acknowledgements
S.L. acknowledges support from UCL's Graduate Research Scholarship and Overseas Research Scholarship. The authors acknowledge support from the EPSRC under grants EP/W026872/1, EP/S005021/1, and thank Andrew Hallam for very helpful discussions.

## Author contributions
A.G.G. conceived the project. S.L. developed the theoretical framework. Both authors contributed to the writing and approved the final manuscript.

## Competing interests
The authors declare no competing interests.
