## [Transparent Peer Review file · Nature Communications]

Entanglement growth from squeezing on the MPS manifold

Corresponding Author: Mr Sebastian Leontica

Version 0:

Reviewer comments:

Reviewer #1

(Remarks to the Author)

This work is a contribution to the varied literature on the somewhat thorny topic of classical-quantum correspondence and especially the genesis of classical chaos from quantum mechanics.

The authors have considered what seems a novel approach, but has a background also from their own previous numerical explorations. Namely they project many-body states onto manifolds of MPS (matrix product states), in this case constructing a so-called Husimi function. The dynamics on the projected manifold is Hamiltonian and this is the classical motion that is associated with the quantum many-body one. This has its roots in the TDVP being applied with MPS. A previous paper (Ref. 49) had explored mainly numerically the Lyapunov spectrum from such a classical dynamics for Hamiltonians that were integrable or chaotic. In this work an analytic connection is made between the growth of the Lyapunov spectrum or the KS entropy and the entanglement. There is a squeezing of a wavepacket in the MPS manifold once the entanglement is saturated on it. This is my rough understanding of the work and let me also say that I have only an acquaintance with TDVP and MPS, not having used them in my research yet. I agree that this approach may be a fruitful one in approaching many-body quantum systems, and perhaps issues like quantum scars may be usefully studied with it. However, I am skeptical for the reasons I include in the following

Comments/Queries:

1. Why do integrable systems have positive Lyapunov spectra as obtained from TDVP-MPS? Classical Integrable systems can be unstable perhaps, but not chaotic. They are not unstable on a nonzero measure set of orbits. Given this I am not sure that the obtained Lyapunov exponents have much to do with a putative semiclassical limit. This may be studied fruitfully if there are models studied which do have classical limits (Bose-Hubbard perhaps, or FPU).

2. Integrable systems can also show linear entanglement growth, may not saturate at the Page value, but can grow at this rate, so the question is associated with the previous one, would one associate with a KS entropy to them?

3. There have been other attempts at deriving some kind of classical flows from the quantum, for example the Bohmian mechanics, or the semiclassical approach, where too one can find chaos in perfectly innocent one-degree of freedom systems.

4.

====

Reviewer #2

(Remarks to the Author)

In this work the Authors work towards a theoretical connection between the growth of entanglement entropy and the Lyapunov spectrum of the projected dynamics onto a variational manifold. Specifically the Authors define and MPS-Husimi distribution and introduce a path-integral approach to bound the entanglement entropy through a combination of quantum and classical entropies. In particular this allows one to attempt to capture the entanglement growth past the saturation of the

MPS entanglement.

While the work itself is rather technical I find that the Authors have nevertheless managed to present it in a rather readable form. That said, I have two questions for the Authors.

1) After Eq. (31) the Authors comment that one can find an approximate solvable model for the dynamics of wave-packets on the MPS manifold, and reference a past work for details. However, at no point is the effect of this approximation discussed. Perhaps the Authors could share some intuition about this and how this approximation could affect their method, if this is understood.

2) On a somewhat similar note, and perhaps this is simply a case of my misunderstanding, but soon after, before Eq. (33) the Authors claim that squeezing is not observed when considering only the regular TDVP equations. Firstly, here one could misunderstand the statement when considering the context in which the classical TDVP equations have been studied in recent years, where the discussion was focused on the dynamics within the space of MPS parameters. In that space the equations are non-linear and often show classically mixed phase spaces, which could naturally lead to deformations of any distributions one would consider over the MPS parameters. Perhaps this could be clarified slightly in the manuscript. Secondly, the Authors then extend these equations to the second order in order to allow for the capture of the squeezing dynamics. However, similarly to before there is no mention of the effects of this choice. How would one expect the results to change as one takes higher orders, or at which time scales? I understand that deriving such equations would be highly non-trivial, but perhaps the Authors have some insight into what they expect that could prove relevant.

Perhaps as a final remark, in Sec. II the Authors present schematics with the tensor equations, which can help some readers. Perhaps such schematics could be added to some of the other Equations (e.g. (11), (12), (16)). This might make the otherwise fairly technical manuscript slightly easier to read, particularly given the audience of the chosen journal.

All in all I find the presented work here to be very interesting as it could lead to further results connecting classical and quantum chaos, which currently have fairly disconnected definitions. Indeed such a connection has long been sought and this work offers a potential new avenue to explore in this direction.

To conclude, I believe that this manuscript is of sufficient quality and relevance for publication in Nature Communications.

Version 1:

Reviewer comments:

Reviewer #2

(Remarks to the Author)

I believe the Authors have addressed all concerns and suggestions of the Referees. Given that these were mostly minor and that the work is of substantially quality and interest, I recommend it for publication in Nature Communications.

Response NCOMMS-24-23484-T

Response to Referee 1:

We thank the referee for their careful reading of our manuscript and, in particular, for pointing out the consequences of our results when applied to integrable systems. While the exact connection between classical and quantum chaos is far from resolved, we believe our approach is a natural middle ground where important progress can be achieved.

Comment 1: Why do integrable systems have positive Lyapunov spectra as obtained from TDVP-MPS? Classical Integrable systems can be unstable perhaps, but not chaotic. They are not unstable on a nonzero measure set of orbits. Given this I am not sure that the obtained Lyapunov exponents have much to do with a putative semiclassical limit. This may be studied fruitfully if there are models studied which do have classical limits (Bose-Hubbard perhaps, or FPU).

Response: We assume this refers to the positive Lyapunov spectrum found for the quantum integrable system discussed in the previous numerical work. At the current time we believe the following resolution exists for this case: the projection of a quantum integrable system onto MPS manifolds (including the $D=1$ standard classical limit) can either be classically integrable or classically chaotic. If the projection is also classically integrable, the entanglement entropy of the quantum system should follow a periodic pattern and not exhibit asymptotic growth. This was analytically shown to be the case for simple coupled quantum harmonic oscillators in 1D. If the projected trajectories are 'almost' stable, as was seen in systems with mixed phase spaces [Phys. Rev. X **10**, 011055], the entanglement entropy indeed grows at a strongly suppressed rate, as expected from our result. When working at higher bond dimension ($D=2$ and above) we showed that an exact account of the local divergence of trajectories must use the geometric correction discussed in this work, which had been missed in previous numerics. In the cases of classically chaotic trajectories, the 'on-average' effect of geometry was argued here to result in a simple $\sim D^2$ factor between the measured KS entropy and entanglement growth, but for periodic orbits this simple averaging is unjustified. We are not able to account for this at a granular level yet and will look into ways of doing this in the future, but we do expect to see a null KS entropy at all bond dimensions for quantum integrable systems with classically integrable counterparts.

Comment 2: Integrable systems can also show linear entanglement growth, may not saturate at the Page value, but can grow at this rate, so the question is associated with the previous one, would one associate with a KS entropy to them?

Response: For such quantum integrable systems with linear growth of entanglement entropy we expect based on our result there should not be a classically integrable counterpart. The projected dynamics should either be classically chaotic, or display some kind of mixed phase space with only approximately periodic orbits. Since the manifold is an overcomplete basis, any state can be represented as a weighted sum of states on it. When entanglement exceeds that of any point on the manifold during a quench, the excess must be captured by the spreading of the Husimi distribution. It would be fascinating to check whether some kind of bridge of intermediate behaviours forms as one projects to manifolds

of higher and higher bond dimensions, but this is another question best treated once some more technical aspects are resolved and we will get back to it in the future.

We believe that the case of integrable systems provides a revealing edge case for our approach. We have added a paragraph discussing these issues in the 'Discussion' section of our revised manuscript. This explains our understanding of this situation, as well as the points made in response to comment 1 above: "An interesting avenue for further investigation [...] suppressed entanglement growth rate."

Comment 3: There have been other attempts at deriving some kind of classical flows from the quantum, for example the Bohmian mechanics, or the semiquantum approach, where too one can find chaos in perfectly innocent one-degree of freedom systems.

Response: The classical dynamics we obtain differ strongly from the other approaches mentioned, in that we do not assume the existence of additional physics beyond unitary evolution and projective measurements. The density matrix corresponding to the Husimi distribution on the manifold is explicitly the result of a physical process, namely the measurement of the quantum state of interest in a basis formed of densely distributed MPS. This means the classical flows described could in principle occur organically, under localizing noise of various strengths that tends to project the system to area-law states, as seen in recently investigated monitored random circuits []. We added a sentence in the 'Discussion' section to clarify this "We note that classical flows [...] entanglement." Considering projections of quantum systems to product states has been very fruitful, so we believe projections to quasi-local states are natural extensions.

We again thank the referee for their careful reading of our manuscript. We hope that, with these comments and with our revisions to the manuscript, the referee will find our work suitable for publication in Nature Communications.

Response to **Referee 2:**

We thank the referee for their careful and positive reading our manuscript and for providing helpful suggestions.

Comment 1: After Eq. (31) the Authors comment that one can find an approximate solvable model for the dynamics of wave-packets on the MPS manifold, and reference a past work for details. However, at no point is the effect of this approximation discussed. Perhaps the Authors could share some intuition about this and how this approximation could affect their method, if this is understood.

Response: We added a paragraph in the "Discussion" section to summarise our understanding of the effect of this approximation: " In the expansion of the Hamiltonian, we neglect [...] towards perturbative corrections.". This also covers the related second part of the follow-up comment below.

Comment 2: On a somewhat similar note, and perhaps this is simply a case of my misunderstanding, but soon after, before Eq. (33) the Authors claim that squeezing is not observed when considering only the regular TDVP equations. Firstly, here one could misunderstand the statement when considering the context in which the classical TDVP equations have been studied in recent years, where the discussion was focused on the dynamics within the space of MPS parameters. In that space the equations are non-linear and often show classically mixed phase spaces, which could naturally lead to deformations of any distributions one would consider over the MPS parameters. Perhaps this could be clarified slightly in the manuscript. Secondly, the Authors then extend these equations to the

second order in order to allow for the capture of the squeezing dynamics. However, similarly to before these is no mention of the effects of this choice. How would one expect the results to change as one takes higher orders, or at which time scales? I understand that deriving such equations would be highly non-trivial, but perhaps the Authors have some insight into what they expect that could prove relevant.

Response: We clarified what we meant by the statement in relation to the goal we are trying to achieve in the space immediately following the remark: " Looking at the squeezing of classical distributions following the TDVP equation is a good starting point, but this leads to the overcounting effect mentioned early when applied to the MPS-Husimi distribution."

Comment 3: Perhaps as a final remark, in Sec. II the Authors present schematics with the tensor equations, which can help some readers. Perhaps such schematics could be added to some of the other Equations (e.g. (11), (12), (16)). This might make the otherwise fairly technical manuscript slightly easier to read, particularly given the audience of the chosen journal.

Response: We added a tensor diagram to Eq. (11). Additionally, Section III was rewritten to omit some of the technical detail (which is now deferred to the Supplemental Material) in favour of a clearer and more intuitive exposition of the MPS-Husimi distribution. We introduced a tensor diagram into Eq. (12), the definition of the MPS-Husimi distribution, to clarify its meaning.

We thank the referee once again for their reading of our manuscript. We hope that they will agree that our response to their comments have made the manuscript more clear.

Summary

- Discussion section was expanded to share some insight about the relevance of our result for the case of integrable systems.
- We justify looking at the projected dynamics as a naturally occurring physical process in the discussion.
- We add to the discussion a paragraph about the effect of truncating the equation of motion of the MPS-Husimi distribution to second order.
- In Section VI, we clarify the distinction between the squeezing of classical distributions due to TDVP equation and the squeezing of the MPS-Husimi distribution.
- Section III was rewritten to provide a clear and intuitive introduction of the MPS-Husimi distribution, with the underlying details moved to the Supplemental Material.
- We added tensor diagrams to Eq. (11) and Eq. (12) for improved readability.